# Breast Cancer Surgery 10-Year Survival Prediction by Machine Learning: A Large Prospective Cohort Study

**DOI:** 10.3390/biology11010047

**Published:** 2021-12-29

**Authors:** Shi-Jer Lou, Ming-Feng Hou, Hong-Tai Chang, Hao-Hsien Lee, Chong-Chi Chiu, Shu-Chuan Jennifer Yeh, Hon-Yi Shi

**Affiliations:** 1Graduate Institute of Technological and Vocational Education, National Pingtung University of Science and Technology, Pingtung 91201, Taiwan; lou@mail.npust.edu.tw; 2Department of Healthcare Administration and Medical Informatics, Kaohsiung Medical University, Kaohsiung 80708, Taiwan; syeh@faculty.nsysu.edu.tw; 3Department of Biomedical Science and Environmental Biology, College of Life Science, Kaohsiung Medical University, Kaohsiung 80708, Taiwan; mifeho@kmu.edu.tw; 4Department of Surgery, Division of Breast Oncology and Surgery, Kaohsiung Medical University Hospital, Kaohsiung 80708, Taiwan; 5Center for Liquid Biopsy and Cohort Research, Kaohsiung Medical University, Kaohsiung 80708, Taiwan; 6Department of Surgery, Kaohsiung Municipal United Hospital, Kaohsiung 80457, Taiwan; hongtchang@gmail.com; 7Department of General Surgery, Chi Mei Medical Center, Liouying 73658, Taiwan; Hao_Hsien@hotmail.com; 8Department of General Surgery, E-Da Cancer Hospital, Kaohsiung 82445, Taiwan; chiuchongchi@yahoo.com.tw; 9School of Medicine, College of Medicine, I-Shou University, Kaohsiung 82445, Taiwan; 10Department of Business Management, National Sun Yat-sen University, Kaohsiung 80420, Taiwan; 11Department of Medical Research, Kaohsiung Medical University Hospital, Kaohsiung 80708, Taiwan; 12Department of Medical Research, China Medical University Hospital, China Medical University, Taichung 40402, Taiwan

**Keywords:** breast cancer surgery, 10-year survival, machine learning, deep neural network, performance

## Abstract

**Simple Summary:**

This study provided an analysis of machine-learning algorithms and the ability to predict 10-year survival after breast cancer surgery. The univariate analyses and the global sensitivity analysis provided in this study are especially helpful. This represents a novel opportunity for understanding the significance of a preoperative SF-36 PCS score, a preoperative SF-36 MCS score, postoperative recurrence, and tumor stage in predicting 10-year survival after breast cancer surgery and could lead to clinicians being better informed about the precision and efficacy of management for these patients. These results encourage a broader international validation of language models in clinical practice and emphasize that preoperative physical and mental functioning should always be an integral part of cancer care. Future studies may investigate further refinements of the machine-learning algorithms applied in this study and their potential for integration with other clinical decision-making tools.

**Abstract:**

Machine learning algorithms have proven to be effective for predicting survival after surgery, but their use for predicting 10-year survival after breast cancer surgery has not yet been discussed. This study compares the accuracy of predicting 10-year survival after breast cancer surgery in the following five models: a deep neural network (DNN), K nearest neighbor (KNN), support vector machine (SVM), naive Bayes classifier (NBC) and Cox regression (COX), and to optimize the weighting of significant predictors. The subjects recruited for this study were breast cancer patients who had received breast cancer surgery (ICD-9 cm 174–174.9) at one of three southern Taiwan medical centers during the 3-year period from June 2007, to June 2010. The registry data for the patients were randomly allocated to three datasets, one for training (*n* = 824), one for testing (*n* = 177), and one for validation (*n* = 177). Prediction performance comparisons revealed that all performance indices for the DNN model were significantly (*p* < 0.001) higher than in the other forecasting models. Notably, the best predictor of 10-year survival after breast cancer surgery was the preoperative Physical Component Summary score on the SF-36. The next best predictors were the preoperative Mental Component Summary score on the SF-36, postoperative recurrence, and tumor stage. The deep-learning DNN model is the most clinically useful method to predict and to identify risk factors for 10-year survival after breast cancer surgery. Future research should explore designs for two-level or multi-level models that provide information on the contextual effects of the risk factors on breast cancer survival.

## 1. Introduction

Breast cancer is the most common cancer diagnosis worldwide and the second most common cause of cancer-related death in women worldwide [1]. In the general population, the rate of survival for breast cancer surgery is high, but various factors can reduce survival substantially, including demographic and clinical characteristics, care quality, and quality of life (QOL) before surgery [2]. Therefore, the ability to obtain accurate predictions of 10-year survival after breast cancer surgery can improve the efficacy of healthcare institutions in allocating, coordinating, and expending limited healthcare resources for treating these patients.

Researchers have developed various models for predicting breast cancer surgery outcomes, but proposed models for predicting survival 10 years after breast cancer surgery consistently reveal three major weaknesses. First, the accuracy of recently proposed models for predicting breast cancer surgery survival is consistently inferior to that of conventional models [3,4]; second, health insurance claims data is the most commonly used input data for the proposed forecasting models is, which have limited real-time availability in the typical clinical scenario [5,6]; and third, most proposed models do not consider factors that have well established associations with breast cancer survival, e.g., demographic and clinical characteristics, care quality, and QOL before surgery [7,8]. Statistical machine learning and deep learning algorithms have been found to have diverse applications in the medical field [4,5,6,7,8,9]. For example, these methods can be used to account for specific clinical and genetic characteristics of the individual patient with a given disease, by improving the accuracy of identifying and ranking risk factors for death from the disease. The continuing accumulation of detailed real-world medical data in the current “information age” and advances in machine learning technologies are providing researchers and practitioners with the ability to generate models that consider numerous predictors in breast cancer mortality risk stratification. The related studies are summarised in Table 1.

The primary aim of this study was to compare five forecasting models in terms of their accuracy in predicting survival within the 10 years following surgery for breast cancer. The five forecasting models include deep neural networks (DNN), K nearest neighbor (KNN), support vector machine (SVM), naive Bayes classifier (NBC) and Cox regression (COX). The secondary aim was to identify significant predictors of survival in the 10 years following breast cancer surgery. The model performance comparison results and the identification of significant predictors of survival have two potential applications, as healthcare administrators and researchers can use the results not only to develop, evaluate, and improve healthcare policies but also to improve healthcare decision making.

## 2. Materials and Methods

### 2.1. Design of Study and Participants

The participants in this prospective cohort study were interviewed using structured questionnaires. The inclusion criteria were a primary diagnostic code for breast cancer (ICD-9 cm 174–174.9) and documentation of breast cancer surgery received at one of three medical centers located in southern Taiwan in the period from June 2007, to June 2010. The following four inclusion criteria were also applied: (1) a record of only one previous breast cancer surgery; (2) a record of breast conservation surgery, modified reconstructive mastectomy, or mastectomy with reconstruction; (3) a clear consciousness and fluency in Chinese or Taiwanese; and (4) consent to be interviewed by the researchers. Four exclusion criteria were applied: (1) the presence of a benign tumor; (2) re-recurrence; (3) cognitive impairment; and (4) refusal to participate. After the application of the above criteria, 1178 of the remaining patients who consented to participate in writing and who completed the SF-36 survey before surgery were enrolled in the study. Figure 1 presents a flowchart of the procedure used to recruit the participants. The institutional review board at Kaohsiung Medical University Hospital (KMUH-IRB-960186) approved the study protocol.

### 2.2. Five Forecasting Models

This study compared the forecasting performance in five models. The first forecasting DNN model used is a simple multilayer perceptron, which contains 4 hidden layers. The sizes of the first two layers were selected as 64 and 64 during hyperparameter tuning. Batch normalization and dropout were also performed after the first two hidden layers [10,11]. Batch normalization was performed to normalize the output that passed into the next layer, as it helps to reduce the covariance shifts of the hidden values. Dropout is used after batch normalization for further regularization. The activation function used for all layers is the Rectified Linear Unit (ReLU). The ReLU function is defined as y = max (0, x), a non-linear function that allows the model to capture more complex relationships. The final activation of the output uses a sigmoid function to produce values between 0 and 1. Additionally, the optimal hyperparameters and architecture for the deep learning DNN model were obtained through grid-search in a hyperparameter search and the number of epochs were decided through the tuning process described in Table 2. The second forecasting model used in this study was the KNN algorithm, in which variables are classified according to the closest training data in the feature space [12]. To perform a majority vote on outcomes of the points that are k-nearest to the new sample, the KNN model uses a simple data mining algorithm, an instance-based learning method. The third forecasting model used in this study was SVM, which is a supervised algorithm that divides the feature space into hyperplanes according to the target classes [13]. The SVM also uses kernel functions to discriminate between nonlinearly separable classes. The fourth model was NBC, which assumes that the presence of a particular feature in a class is unrelated to the presence of any other feature [14]. That is, each feature is considered to be an independent and equal contributor to the outcome. The fifth forecasting model was the COX model, which is essentially a proportional hazards regression model. The COX model is a widely used statistical tool in medical research for predicting patient survival, i.e., for investigating whether patient survival is associated with one or more variables [15].

### 2.3. Potential Predictors

Patient data retrieved from patient medical records included demographic characteristics (years of age, years of education, current residence with other family members, marital status, body mass index, Charlson comorbidity index, size of tumor, stage of tumor, use of tobacco, use of alcohol, and history of breast cancer), clinical characteristics (surgery type, American Society of Anesthesiologists score, chemotherapy, radiotherapy, and hormonal therapy), care quality (length of hospital stay after surgery, rehospitalization within 30 following after surgery, cancer recurrence, survival, and reconstructive surgery), and preoperative QOL (preoperative SF-36 Physical Component Summary (PCS) score and Mental Component Summary (MCS) score). This study used the Chinese version of the SF-36. The Chinese version is well-validated and is commonly used in both clinical practice and research [16]. To assess the overall physical functioning and mental functioning in the study population in comparison with the general population of Taiwan, norm-based scoring methods were used to calculate SF-36 PCS and MCS scores. A procedure for converting the SF-36 PCS scores and MCS scores was performed to obtain the mean of 50, and standard deviations of 10 in comparison with a “normal” group of breast cancer surgery patients drawn from a nationwide population [17]. Multivariate analyses were performed with the potential predictors as independent variables and survival 10 years after breast cancer surgery was used as the dependent variable. Additionally, several data pre-processing methods were conducted in preparation of the development of the prognostic models. Missing data creates difficulties for the development of machine-learning models. As the 10-year survival after breast cancer surgery was an outcome of the study, patients with missing data on 10-year survival were excluded when developing the machine-learning models for that outcome.

### 2.4. Statistical Analysis

The individual patient who had received surgery for breast cancer was used as a unit of analysis. The four steps of the statistical analysis in this study were as follows. First, the 1178 cases in the overall database were randomized into a dataset of training (824 cases) for use in model development, a dataset of testing (177 cases) for use in internal validation, and a dataset of validating (177 cases) for use in external validation. Next, the independent variables (significant predictors) and the dependent variable (10-year survival) were fitted to the forecasting models. After model training, model outputs were collected for each testing dataset. The second step of the statistical analysis was performing univariate Cox regression analyses to identify significant (*p* < 0.05) predictors of 10-year survival. To compare the study characteristics between the training dataset and the testing dataset, a one-way analysis of variance was used to determine the statistical significance of continuous variables, and a Fisher exact analysis was used to determine the statistical significance of categorical variables (*p* < 0.05). The third step of the statistical analysis consisted of comparing 1000 pairs of forecasting models with 95% confidence intervals in terms of their accuracy in predicting survival in the 10 years following breast cancer surgery. An independent t test was used to determine whether performance indices significantly differed between each pair of models. Model performance was compared in terms of sensitivity, specificity, positive predictive value (PPV), negative predictive value (NPV), accuracy, and area under the receiver operating characteristics (AUROC) curve. In the last step of the statistical analysis, we performed a global sensitivity analysis to identify variables that were significant predictors of survival. The global sensitivity analysis was employed to identify the most influential parameters, and the input variables against the output variable was expressed as the ratio of the network error (sum of squared residuals). If a variable had a variable sensitivity ratio (VSR) that was equal to or lower than 1, the variable was assumed to diminish performance and was removed.

The scikit-learn 0.21.2 function in Python (v3.7.6; Python Software Foundation, Wilmington, DE, USA) was used to run the deep-learning DNN and other machine learning models, and the Cox proportional hazard model was computed with the Lifelines v0.22.2 function in Python v3.7.6 and double-checked with JMP10.0 (SAS Institute Inc., Cary, NC, USA). All statistical tests were two-sided; a *p* value of less than 0.05 was considered statistically significant.

## 3. Results

### 3.1. Study Characteristics

Table 3 shows that the mean age of the patients who had undergone surgery for breast cancer was 52.2 years (standard deviation, 11.1 years). The tumor stage in the largest proportion (37.4%) of patients was tumor stage II. Most of the breast cancer patients (881 patients, 74.8%) had survived 10 years after surgery. Table 4 presents the univariate Cox regression analysis results, which reveals that 10-year survival after breast cancer surgery was significantly associated with the demographic and clinical characteristics of the patient, the quality of care received in the 10 years following surgery, and with QOL before surgery (i.e., preoperative SF-36 PCS and MCS scores) (*p* < 0.05). Therefore, these predictors were included in the forecasting models.

### 3.2. Comparison of Forecasting Models

As Table 5 indicates, the dataset of training and the dataset of testing did not significantly differ in study characteristics, including 10-year survival after breast cancer surgery; therefore, samples were compared between the two datasets to increase the reliability of the validation results. The data in Figure 2 also indicate that the DNN model compared to KNN, SVM, NBC, COX models had a significantly (*p* < 0.001) higher sensitivity (97.18%, 47.37%, 73.24%, 41.41%, 78.87%, respectively), specificity (98.12%, 92.31%, 96.71%, 90.00%, 73.29%, respectively), PPV (94.52%, 69.23%, 88.14%, 100.00%, 21.37%, respectively), NPV (99.05%, 82.76%, 91.56%, 75.27%, 31.82%, respectively), prediction accuracy (97.89%, 80.28%, 90.85%, 75.35%, 22.18%, respectively), and AUROC (99.70%, 70.00%, 85.00%, 50.00%, 41.10%, respectively) values. Similar results were also observed in the dataset for testing and dataset for validating simultaneously.

### 3.3. Significant Predictors in the DNN Model

To identify the best predictors of survival, the training dataset was used to calculate VSRs for the DNN model. Figure 3 presents the global sensitivity analysis results. As a predictor of 10-year survival after breast cancer surgery, the preoperative SF-36 PCS score had the highest VSR (6.61), followed by the preoperative SF-36 MCS score (VSR = 5.18), postoperative recurrence (VSR = 3.05), and tumor stage (VSR = 1.58). All predictors in the DNN models had VSR values of higher than 1. Therefore, all variables improved the prediction performance of the DNN model.

### 3.4. Sensitivity Analysis

A further 177 validating datasets were used to verify the predictive accuracy in the five models. Figure 2 compares the performance indices obtained in the external validation of the models. With regard to predicting survival in the 10 years following breast cancer surgery, all performance indices for the DNN model were again superior to those for the other forecasting models (*p* < 0.001).

## 4. Discussion

This study provided an analysis of machine learning algorithms and the ability to predict 10-year survival after breast cancer surgery, and the univariate analyses and the global sensitivity analysis were especially helpful. This represents a novel opportunity for understanding the significance of preoperative SF-36 PCS score, preoperative SF-36 MCS score, postoperative recurrence, and the tumor stage in predicting 10-year survival after breast cancer surgery and could lead to clinicians being better informed regarding the precision and efficacy of management for these patients. Furthermore, according to our recent comprehensive review of the literature on machine learning, this study is apparently the first to report the results of a performance comparison in machine learning algorithms for predicting survival in the 10 years following breast cancer surgery. The prediction performance of the deep-learning DNN model was clearly superior when all five forecasting models were constructed using the same set of clinical inputs. Our survival analysis results can be considered to be relatively reliable because the predictions were based on prospective, longitudinal, and long-term (10-year) data obtained from multiple medical institutions. Compared to the prediction models discussed in previous works, in which predictions were based on a dataset for a single medical center [4,5,6], the use of data from several institutions in our study provides a relatively more accurate and reliable estimate of survival after breast cancer surgery. Additionally, the data used in this study were registry data compiled from data for several institutions. In comparison with the use of data for a single institution, the use of registry data in this study improved accuracy in depicting breast cancer surgery treatment for a large population [4,5,6,18]. Another advantage of using registry data was that the potential effects of a bias resulting from the referral of patients or the bias resulting from the practices of a single high-volume surgeon or a single high-volume institution were minimized [18].

This study had several notable strengths. First, this study is, to the best of our knowledge, the first to compare the performance of machine learning algorithms, including regression-based method, in predicting survival in the 10 years following surgery in a large general population of patients who had received surgery for breast cancer. In contrast with other machine learning tools proposed for prognostic use in oncology, this study performed model training using data for all patients treated at oncology or hematology/oncology clinics. That is, all patients were included regardless of whether they had received cancer-directed therapy [5,7,8]. Another strength of this study is that the forecasting models and machine learning algorithms in this study included a higher number of predictors compared to those reported in the literature, and data for all of the included predictors were typically available in real time and in structured formats from medical recorder databases [6,7,8]. Therefore, in the general oncology setting, model training can be performed more efficiently in the proposed forecasting models compared to previous machine-learning algorithms. A final strength of this prospective longitudinal cohort study is that patients were followed up over a 10-year period, which is longer than the follow up performed in previous works. The long follow-up period was essential, because, in the typical clinical setting, most of the patients that the model classified as high-risk patients would receive counseling in terms end-of-life preferences. Despite the above strengths of this study, the findings should be interpreted cautiously because the gradient-boosting model that was used in this study was an older version with a less robust feature selection and hyperparameter optimization compared to recently developed models.

Compared to the other models, the superior forecasting performance of the DNN model and other advantages resulting from its unique characteristics are well established in the literature and are well supported by comprehensive statistical analyses and comparisons in previous works [19,20,21]. One advantage of the DNN model is its ability to process incomplete or noisy inputs more appropriately and more accurately compared to other models when no missing data occurs in the dataset. Another advantage is that linear and non-linear DNN models, which have many potential applications in analyzing data contained in large-scale medical databases, are easy to construct as long as the input data are highly correlated, even if they are not normally distributed. Predicting prognosis is only one of many potential applications of DNN models in clinical research. The model proposed in this study can also be extended to predicting outcomes of treatments other than for breast cancer surgery.

This study performed a global sensitivity analysis of the weights of significant predictors of 10-year survival in patients who had received breast cancer surgery. The best predictor of survival was the SF-36 PCS score, and the next best predictor was SF-36 MCS score. This finding supports earlier reports that SF-36 PCS and MCS scores are the best predictors of breast cancer surgery outcomes. Specifically, PCS and MCS scores are better outcome predictors in comparison with cost of treatment, QOL, hospital readmission, complications, and overall post-surgery survival [22,23,24]. In a recent prospective cohort study, Chiu et al. performed a longitudinal analysis studying the effect of preoperative QOL on minimal clinically important differences (MCIDs) and survival in patients who had received surgical resection of hepatocellular carcinoma [24]. The authors reported that preoperative SF-36 PCS and MCS scores were significant independent predictors of MCIDs and survival after resection (*p* < 0.001). The most suitable explanation is that patients who already have high QOL scores before surgery has less potential to achieve QOL improvements large enough to meet MCID criteria. Another possible explanation is that the high subjectivity of the QOL score as a measure of physical and emotional impacts of cancer or its treatment makes it a less reliable measure compared to traditionally applied measures, which are relatively more objective. Regarding the use of QOL scores as predictors of cancer survival, Quinten et al. performed a meta-analysis of patient data from a selection of 30 randomized controlled trials to investigate whether baseline QOL is a prognostic predictor of cancer survival [25]. Their meta-analysis, which included data for 10,108 patients with cancer at 11 different sites, revealed that baseline QOL is, in addition to biological measures, a significant independent predictor of survival in the general population of cancer patients. Currently, preoperative SF-36 PCS and MCS scores are well recognized as useful outcome predictors in patients who have undergone cancer surgery. For investigators, the use of these scores provides a more comprehensive depiction of the potential outcomes of a proposed (palliative) treatment, including potential negative outcomes such as reduction in functional status and reduction in overall QOL. Thus, in addition to considering clinical outcomes, future randomized controlled trials should consider QOL as a standard outcome measure. Stratifying patients by baseline QOL in future trials would increase the homogeneity of treatment groups, which would then improve the reliability of the results and simplify the interpretation of the results.

This study revealed a significant negative association between the recurrence of cancer after breast cancer surgery and 10-year survival after surgery. During the study period, 219 patients (18.6%) suffered postoperative recurrence, and the regular surveillance for cancer recurrence is known to be an independent protective factor in cancer survival [26,27]. A patient who undergoes regular surveillance has an improved likelihood of receiving treatment that has curative potential at or near the time of cancer recurrence, which then improves survival. For example, a recent multicenter clinical trial of a large population of patients with hepatitis B virus-related hepatocellular carcinoma reported that cancer recurrence within less than 2 years following curative resection was independently associated with 10-year survival. Curative treatment for the first recurrence of cancer suffered by the patient was identified as another independent protective factor in 10-year survival was [28]. For an improved long-term survival rate of patients who require surgical treatment for cancer, regular surveillance for cancer recurrence after surgery is essential. Therefore, clinicians should aim to provide their patients with sufficient information regarding recurrence, including rate of recurrence, signs and symptoms of recurrence, and practices and interventions for reducing recurrence risk. Additionally, patients are more likely to comply with the scheduled follow up and surveillance procedures if they clearly understand the underlying rationale for such procedures.

The importance of a surveillance program for early diagnosis of cancer recurrence was well established in a recent retrospective study by Lee et al. The authors analyzed patterns of recurrence in patients who had received curative hepatectomy for hepatocellular carcinoma and discussed the implications of recurrence patterns for postoperative surveillance [29]. The authors concluded that, in patients who underwent curative hepatectomy for hepatocellular carcinoma, recurrence was very common; therefore, the early diagnosis of hepatocellular carcinoma recurrence and early curative retreatment can improve survival in these patients. After surgery, breast cancer patients are vulnerable to various cancer-related comorbidities that can contribute to poor outcomes of surgery, e.g., postoperative complications, extended hospital stay, short survival, and high cost of treatment.

Finally, surveillance is important for detecting cancer at an early stage, the point at which the widest range of treatment options is still available and when the chances of survival and recovery are relatively high. For example, Chou et al. reported that survival after cancer surgery decreases as tumor stage increases [30]. Our global sensitivity analysis also indicated that postoperative 10-year survival tends to decrease in patients with late-stage tumors, which is consistent with other studies [30,31].

For a further validation of the significant association observed between risk factors and 10-year survival after breast cancer surgery, Table 6 lists selected studies that have identified risk factors for poor survival after breast cancer surgery [24,30,32,33,34,35,36]. As in these previous works, our study demonstrated that the preoperative SF-36 PCS score, preoperative SF-36 MCS score, postoperative recurrence, and tumor stage are significantly associated with 10-year survival after breast cancer surgery (*p* < 0.05).

This prospective observational study investigated the survival outcomes in a cohort of breast cancer patients who had undergone breast cancer surgery at one of several healthcare institutions in Taiwan. The deep-learning DNN model developed in this study accurately identified factors significantly associated with survival within 10 years after surgery. However, the proposed forecasting model has many possible clinical applications other than prediction of survival after surgery. For example, one potential application by healthcare institutions is in evaluating the effectiveness of medical treatment, which is essential not only for maintaining and improving the quality of healthcare, but also for reducing healthcare costs and for the efficient allocation of limited healthcare resources. Since the proposed DNN model demonstrated satisfactory accuracy in predicting survival in the 10 years following a breast-cancer-surgery procedure, performed in one of the participating institutions, healthcare administrators at other institutions can use the model to demonstrate the need for prompt and appropriate postsurgical treatment. Broader potential applications of the model in Taiwan and elsewhere include the development and promotion of public healthcare policies as well as the development of decision-support systems, which would ultimately contribute to improved health and outcomes, not only in breast cancer surgery patients, but in all cancer patients. Although the results of this study indicate that the DNN model has a strong potential application in the healthcare field, further studies are needed to determine the true clinical relevance of the DNN model, and to clarify its practical clinical applications in predicting prognosis and in optimizing medical management for breast cancer patients after surgery.

Since the results of this study were derived through an analysis of a large database, some limitations should be considered when interpreting the results and applying them in practice. First, our study revealed the numbers of modified radical mastectomy or mastectomy with reconstruction being higher than that of breast-conserving therapy, which is contradictory to the prevalence in the USA and Europe. The process of making a treatment decision is complicated and involves many factors influencing patients’ choice of surgery type, and therefore requires further study. Second, the comparisons made in this study do not consider post-surgery complications that are known to be associated with poor survival after breast cancer surgery. Third, although the datasets used include several variables, it lacks some of the key variables for predicting 10 year survival. These include the intrinsic subtype, pathological factors, multi-gene assay, etc. Fourth, the comparisons were limited to individual DNN, KNN, SVM, NBC and COX models. Future works may consider using an alternative study design that compares a balanced sample of preoperative SF-36 PCS or MCS scores at the first level and then randomly selects breast cancer patients at the second level. Multilevel modeling may also be useful for detecting the interactive effects of patient characteristics, clinical characteristics, quality of care and preoperative QOL in breast cancer patients who suffer recurrence. Finally, further studies are needed to compare performance among different combinations of forecasting models, particularly in analysis of medical data. Despite the limitations acknowledged above, the robustness and statistical significance of the results obtained in this study support the validity of its conclusions.

## 5. Conclusions

The results of the model-performance comparisons in this study support our conclusion that the deep-learning DNN model is the most clinically useful method in predicting survival in the 10 years following surgery for breast cancer. For breast cancer patients who are candidates for breast cancer surgery or have already received surgery, the survival predictors identified in this study can be used to educate patients in terms of the likely course of recovery after surgery and other health outcomes. The results of the current study further suggest that 10-year survival among women with breast cancer surgery could be enhanced by targeted interventions aimed at increasing patients’ overall physical and mental functioning. The implications of these findings can be profound, as surgeons and patients can be equipped with a method to predict 10-year survival after surgery. The current study suggests that cancer survival may be improved through preoperative physical and mental functioning, and that there is always time for such methods to be implemented. These results encourage a broader international validation of language models in clinical practice and emphasize that preoperative physical and mental functioning should always be an integral part of cancer care. Future studies may investigate further refinements of the machine-learning algorithms applied in this study and their potential integration with other clinical decision-making tools. Hybrid methods may provide additional data that can be used to improve the prediction of survival after breast cancer surgery. Such data could also be vital for developing, promoting, and improving healthcare policies related to post-surgery treatment of breast cancer patients. Additionally, future research can explore designs for two-level or multi-level models that provide information on the contextual effects of preoperative SF-36 PCS and MCS scores on breast cancer survival.

## Figures and Tables

**Figure 1 biology-11-00047-f001:**
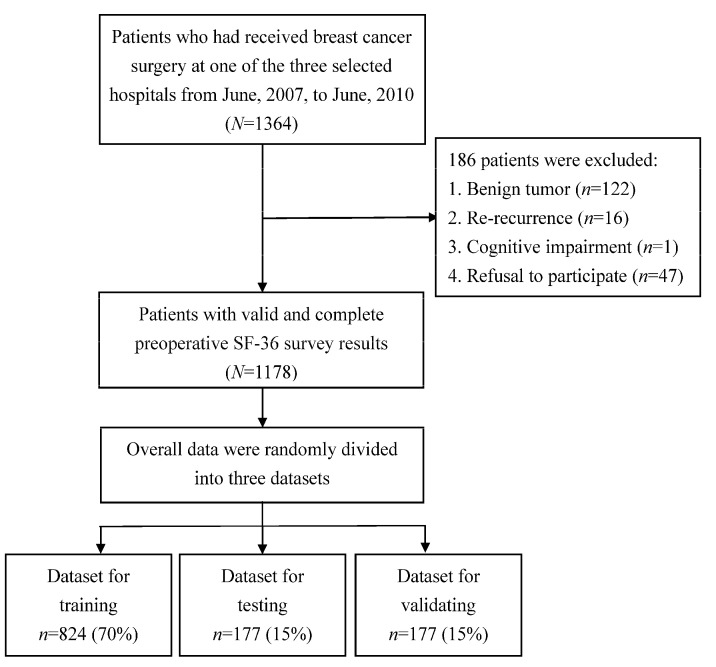
Flowchart of the study procedure.

**Figure 2 biology-11-00047-f002:**
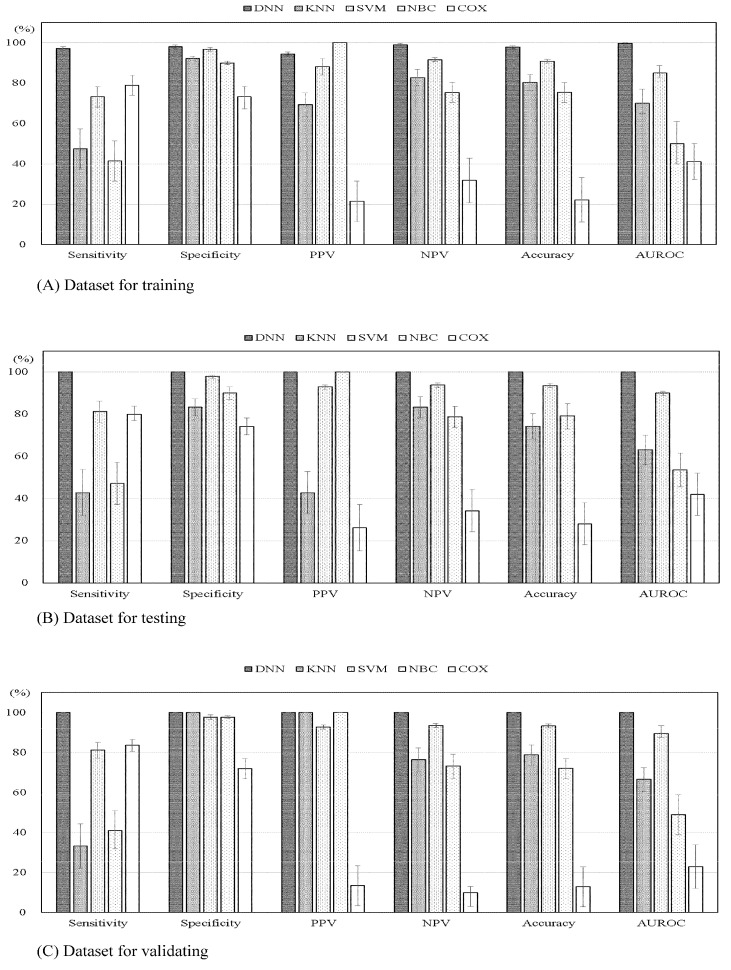
Machine learning model comparison in terms of performance indices with 95% confidence intervals for predicting 10-year survival after breast cancer surgery. (**A**) Dataset for training. All *p* values < 0.001. (**B**) Dataset for testing. All *p* values < 0.001. (**C**) Dataset for validating. All *p* values < 0.001. Abbreviation: DNN, deep neural networks; KNN, k-nearest neighbor; SVM, support vector machine; NBC, naïve Bayesian classifier; PPV, positive predictive value; NPV, negative predictive value; AUROC, area under the receiver-operating characteristic curve.

**Figure 3 biology-11-00047-f003:**
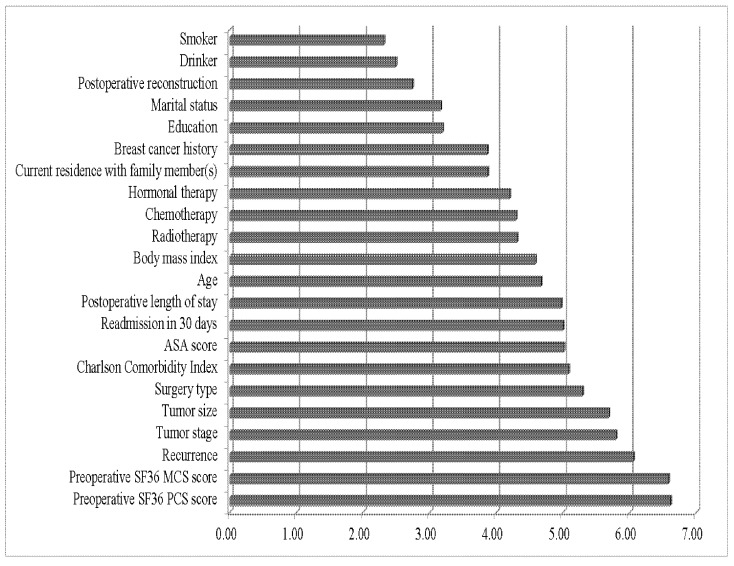
Global sensitivity analysis of deep neural network model in predicting 10-year survival after breast cancer surgery (*N* = 824). Abbreviation: VSR, variable sensitivity ratio; ASA, American Society of Anesthesiologists; PCS, physical component summary; MCS, mental component summary.

**Table 1 biology-11-00047-t001:** Related work summary.

Authors (Years)	Study Sample (Data)	Forecasting Models
Moncada-Torres et al., (2021) [4]	36,658 non-metastatic breast cancer patients from the Netherlands Cancer Registry (NCR) dataset	Random survival forests (RF), survival support vector machines (SVM), extreme gradient boosting (XGBoost), and Cox proportional hazards (CPH)
Kuruc et al., (2021) [5]	RNA-seq data from the Cancer Genome Atlas (TCGA)	Deep neural networks (DNN), Cox proportional hazards (CPH)
Wang et al., (2021) [6]	1137 patients with IB-IIA stage non-small cell lung cancer (China) and compared generalization performance on the Surveillance, Epidemiology, and End Results Program (SEER) dataset	Deep neural networks (DNN), Cox proportional hazards (CPH), SurvNet
Bhambhvani, et al., (2021) [7]	277 patients with genitourinary rhabdomyosarcoma from the Surveillance, Epidemiology, and End Results Program (SEER) dataset	Deep neural networks (DNN), Cox proportional hazards (CPH)
Hou et al., (2020) [8]	7127 breast cancer cases and 7127 matched healthy controls (China)	Extreme gradient boosting (XGBoost), random forest (RF), deep neural network (DNN), logistic regression (LR)

**Table 2 biology-11-00047-t002:** Parameters for classification models.

Parameters	Deep Neural Networks
No. of hidden layers	4
No. of neuron in each hidden layers	(64, 64, 128, 256)
Activation functions in each layer	Rectified linear unit (ReLU) in hidden layers and sigmoid on output layer
Loss function	Binary cross entropy
Optimizer	Adaptive moments estimation (Adam) with 0.001 learning rate
No. of Epochs	100
Dropout layers for regularization	20% dropout layer after second hidden layer and 10% after third hidden layers

**Table 3 biology-11-00047-t003:** Characteristics of patients who had received breast cancer surgery at selected institutions (*N* = 1178).

Variables	N (%)	Mean ± SD
*Demographic characteristics*		
Age, years		52.2 ± 11.1
Education, years		10.2 ± 3.8
Current residence with family member(s)	1127 (95.7%)	
Married	1038 (88.1%)	
Body mass index, kg/m^2^		24.5 ± 4.6
Charlson Comorbidity Index, score		1.0 ± 1.4
Tumor size		2.4 ± 1.8
Tumor stage		
0	80 (6.8%)	
I	354 (30.1%)	
II	441 (37.4%)	
III	303 (25.7%)	
Smoker	55 (4.7%)	
Drinker	29 (2.5%)	
Breast cancer history	150 (12.7%)	
*Clinical characteristics*		
Surgery		
BCS	154 (13.1%)	
MRM	297 (25.2%)	
Mastectomy with reconstruction	727 (61.7%)	
ASA score		2.0 ± 0.4
Chemotherapy	788 (66.9%)	
Radiotherapy	675 (57.3%)	
Hormonal therapy	717 (60.9%)	
*Quality of care*		
Postoperative length of stay, days		2.9 ± 4.7
Readmission in 30 days	283 (24.0%)	
Recurrence	219 (18.6%)	
Survival	881 (74.8%)	
Reconstruction	125 (10.6%)	
*Preoperative quality of life*		
Preoperative SF36 PCS score		56.0 ± 7.6
Preoperative SF36 MCS score		48.8 ± 16.2

Abbreviation: BCS, breast conserving surgery; MRM, modified radical mastectomy; ASA, American Society of Anesthesiologists; PCS, physical component summary; MCS, mental component summary.

**Table 4 biology-11-00047-t004:** Univariate Cox regression analysis of associations between demographic/clinical characteristics of breast cancer patients and survival 10 years after surgery (*N* = 1178).

Variables	HR	*p* Value
*Demographic characteristics*		
Age, years	0.98	<0.001
Education, years	0.90	<0.001
Current residence with family member(s) (no vs. yes)	0.33	<0.001
Marital status (unmarried vs. married)	0.57	<0.001
Body mass index, kg/m^2^	0.96	<0.001
Charlson Comorbidity Index, score	0.81	0.001
Tumor size, cm	0.83	<0.001
Tumor stage		
I vs. 0	0.04	0.001
II vs. 0	0.17	<0.001
≥III vs. 0	0.22	<0.001
Smoker (no vs. yes)	1.36	0.043
Drinker (no vs. yes)	2.09	0.037
Breast cancer history (no vs. yes)	2.70	0.001
*Clinical characteristics*		
Surgery type		
MRM vs. BCS	0.49	0.001
Mastectomy with reconstruction vs. BCS	0.35	<0.001
ASA score	0.35	<0.001
Chemotherapy (no vs. yes)	0.46	<0.001
Radiotherapy (no vs. yes)	0.39	<0.001
Hormonal therapy (no vs. yes)	0.29	<0.001
*Quality of care*		
Postoperative length of stay, days	0.71	<0.001
Readmission in 30 days (no vs. yes)	3.26	<0.001
Recurrence (no vs. yes)	2.17	0.002
Postoperative reconstruction (no vs. yes)	0.39	0.005
*Preoperative quality of life*		
Preoperative SF36 PCS score	1.02	<0.001
Preoperative SF36 MCS score	1.03	<0.001

Abbreviation: HR, hazards ratio; BCS, breast conserving surgery; MRM, modified radical mastectomy; ASA, American Society of Anesthesiologists; PCS, physical component summary; MCS, mental component summary.

**Table 5 biology-11-00047-t005:** Demographic and clinical characteristics of breast cancer surgery patients in training dataset versus testing dataset.

Variables	Training Dataset (*n* = 824)	Testing Dataset (*n* = 177)	*p* Value
*Demographic characteristics*			
Age, years	52.7 ± 10.6	52.5 ± 13.1	0.148
Education, years	10.1 ± 3.8	10.6 ± 4.0	0.174
Current residence with family member(s)	787 (95.5%)	168 (94.9%)	0.900
Married	722 (87.6%)	159 (89.8%)	0.598
Body mass index, kg/m^2^	24.7 ± 5.1	24.0 ± 3.8	0.481
Charlson Comorbidity Index, score	1.0 ± 1.4	1.0 ± 1.3	0.570
Tumor size, cm	2.4 ± 1.9	2.4 ± 1.4	0.344
Tumor stage			0.690
0	67 (8.1%)	7 (3.9%)	
I	251 (30.5%)	57 (32.5%)	
II	305 (37.0%)	67 (37.7%)	
≥III	201 (24.4%)	46 (25.9%)	
Smoker	35 (4.2%)	12 (6.5%)	0.425
Drinker	18 (2.2%)	5 (2.6%)	0.711
Breast cancer history	74 (9.0%)	18 (10.4%)	0.755
*Clinical characteristics*			
Surgery type			0.492
BCS	118 (14.3%)	21 (11.7%)	
MRM	199 (24.1%)	55 (31.2%)	
Mastectomy with reconstruction	507 (61.6%)	101 (57.1%)	
ASA score	2.0 ± 0.4	2.0 ± 0.3	0.676
Chemotherapy	550 (66.7%)	124 (70.1%)	0.572
Radiotherapy	473 (57.4%)	108 (61.0%)	0.565
Hormonal therapy	496 (60.2%)	113 (63.6%)	0.582
*Quality of care*			
Postoperative length of stay, days	2.7 ± 1.9	2.9 ± 1.5	0.711
Readmission in 30 days	185 (22.4%)	48 (27.2%)	0.357
Recurrence	138 (16.8%)	48 (27.2%)	0.067
Postoperative reconstruction Survival	74 (9.0%) 616 (74.8%)	18 (10.4%) 133 (75.3%)	0.564 0.846
*Preoperative quality of life*			
Preoperative SF36 PCS score	56.0 ± 7.6	54.1 ± 6.6	0.758
Preoperative SF36 MCS score	48.4 ± 18.5	49.6 ± 4.2	0.863

Abbreviation: BCS, breast conserving surgery; MRM, modified radical mastectomy; ASA, American Society of Anesthesiologists; PCS, physical component summary; MCS, mental component summary.

**Table 6 biology-11-00047-t006:** Mortality risk factors after breast cancer surgery: factors reported in selected studies.

Authors (Country)	No. of Subjects	Measures	Findings
Chiu et al., 2019 (Taiwan) [24]	369 patients with hepatocellular carcinoma	Functional Assessment of Cancer Therapy-Hepatobiliary (FACT-Hep) and the SF-36	1. Overall postoperative survival was significantly associated with preoperative SF-36 physical component summary score (hazard ratio, HR = 1.05, *p* < 0.001) and preoperative SF-36 mental component summary score (HR = 1.03, *p* < 0.001). 2. Overall postoperative survival was significantly associated with preoperative FACT g total score (HR = 1.07, *p* < 0.001) and preoperative FACT-Hep total score (HR = 1.10, *p* < 0.001).
Quinten et al., 2014 [32]	11 different cancer sites pooled from 30 EORTC randomized controlled trials were selected for this study (7417 cancer patients)	European Organisation for Research and Treatment of Cancer 30-Item Core Quality of Life Questionnaire (EORTC-QLQ-C30)	Overall postoperative survival was significantly associated with preoperative EORTC-QLQ-C30 physical functioning (HR = 0.86, *p* = 0.0119), emotional functioning (HR = 1.13, *p* = 0.002), global health status (HR = 0.92, *p* = 0.017), and nausea and vomiting (HR = 1.17, *p* = 0.002).
Heijl et al., 2010 (Netherlands) [33]	220 patients with potentially curable esophageal adenocarcinoma	Medical Outcome Study Short Form-20 (SF-20) and Rotterdam Symptom Checklist (RSCL)	1. Overall postoperative survival was significantly associated with preoperative SF-20 physical symptom scale (HR = 0.67, *p* = 0.021) and endosonographic T-stage (HR = 0.05, *p* = 0.003). 2. Disease-free postoperative survival was significantly associated with preoperative SF-20 physical symptom scale (HR = 0.64, *p* = 0.024) and endosonographic T-stage (HR = 0.03, *p* < 0.001).
Chen et al., 2020 (China) [34]	149 patients with gastric cancer		Overall postoperative survival was significantly associated with early recurrence during the study period (*p* = 0.011).
Huh et al., 2013 (South Korea) [35]	1159 patients with colorectal cancer		Overall postoperative survival was significantly associated with early postoperative recurrence (HR = 2.42, *p* < 0.001) and tumor stage (HR = 2.38, *p* < 0.001).
Knight et al., 2021 [36]	15,958 patients with colorectal and gastric cancer from 428 hospitals in 82 countries		Overall postoperative survival was significantly associated with cancer stage (odds ratio = 1.80, *p* = 0.036).
Chou et al., 2016 (Taiwan) [30]	8425 patients over 70 years old with solid cancer		3-month postoperative survival was significantly associated with tumor stage (II, III, IV vs. I) (HR = 1.66~4.23, *p* < 0.001).

## Data Availability

The datasets used/or analyzed during the current study are available from the corresponding author on reasonable request.

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
