# Peer review of "Breast Cancer Surgery 10-Year Survival Prediction by Machine Learning: A Large Prospective Cohort Study"

_biology, 2021, doi:10.3390/biology11010047_

Round 1

Reviewer 1 Report

I would like to congratulate for your effort! This study is original and well organized. In my opinion. It should be accepted.

Author Response

I would like to congratulate for your effort! This study is original and well organized. In my opinion. It should be accepted.

Ans: Thank you for your encouraging comments.

Reviewer 2 Report

This paper intends to compare accuracy in predicting 10-year survival after breast cancer surgery of five machine learning models. The work also intends to optimize the weighting of significant predictors.

The subject is interesting and relevant and the article is very well written. I congratulate the authors for their excellent work.

Some topics and excerpts from the text were not clear to me and I believe they could be improved for the final version:

1 -  In section 2.2 (lines 113 and 114) when referring to training the DNN model using transfer learning, the authors mention possible convolutional layers that are freezed during training. I would like to know the original model on which transfer learning was applied. I also got the impression that the DNN model used has only 4 layers of fully connected neurons without any convolutional layer.

2- Despite making the performance comparison of 5 models, only the DNN model had all its hyper-parameters specified in the article. I suggest that all hyper-parameters selected for the other models are also specified, as a deficient selection of hyper-parameters can lead to poor performance of the other classifiers, bringing a bias to the obtained result.

3- The methodology for optimize the weighting of significante predictors should be improved. It was not clear to me how the authors conduct the global sensitivity analysis of deep neural network model and came to the conclusion that the best predictor of 10-year survival after breast cancer surgery was the preoperative Physical Component Summary score on the SF-36. This global sensitivity analysis was also conducted with the other modes for confirmation?

4- With regard to the pre-processing of training data, it was not clear whether there were any missing data in the dataset and how they were treated. Nor did I see any mention of data normalization (linear or z-score?) and the possible handling of outliers.

5- During the Discussions, the authors claims that "One advantage of the DNN model is its capability to process incomplete or noisy inputs". This is not completely true, especially when missing data occurs in the dataset. Other models, such as those based on decision trees (e.g. XGBoost) are more robust against the occurrence of missing data.

6- Finally, I suggest that authors make their data and code publicly available in some repository (e.g. github) ommiting any sensitive information so that other researchers can reproduce the experiments described in the article and can conduct new experiments.

Author Response

Comments and Suggestions for Authors

This paper intends to compare accuracy in predicting 10-year survival after breast cancer surgery of five machine learning models. The work also intends to optimize the weighting of significant predictors.

The subject is interesting and relevant and the article is very well written. I congratulate the authors for their excellent work.

Some topics and excerpts from the text were not clear to me and I believe they could be improved for the final version:

1 -  In section 2.2 (lines 113 and 114) when referring to training the DNN model using transfer learning, the authors mention possible convolutional layers that are freezed during training. I would like to know the original model on which transfer learning was applied. I also got the impression that the DNN model used has only 4 layers of fully connected neurons without any convolutional layer.

Ans: Thank you for your observation. As advised, subsection 2.2 Five Forecasting Models in the revised manuscript provides details of the DNN model as follows: “For 10-year survival after breast cancer surgery, the DNN used is a simple multilayer perceptron which contains 4 hidden layers. The sizes of the first two layers were selected as 64 and 64 during hyperparameter tuning. Batch normalization and dropout were also done after the first two hidden layers [10,11]. Batch normalization was done to normalize the output passed into the next layer, as it helps with reducing the covariance shifts of the hidden values. Dropout is used after batch normalization for further regularisation. The activation function used for all layers is the Rectified Linear Unit (ReLU). The ReLU function is defined as y = max (0, x), a non-linear function that allows the model to capture more complex relationships. The final activation of the output uses a sigmoid function to produce values between 0 and 1.” Additionally, the optimal hyperparameters and architectures for the deep learning DNN model were obtained through grid-search in hyperparameter search and the numbers of epochs were decided through a tuning process described in Table 2.

2- Despite making the performance comparison of 5 models, only the DNN model had all its hyper-parameters specified in the article. I suggest that all hyper-parameters selected for the other models are also specified, as a deficient selection of hyper-parameters can lead to poor performance of the other classifiers, bringing a bias to the obtained result.

Ans: The specific values of the performance indices in whole forecasting models have been added in subsection 3.2 Comparison of Forecasting Models. Again, thank you.

3- The methodology for optimize the weighting of significante predictors should be improved. It was not clear to me how the authors conduct the global sensitivity analysis of deep neural network model and came to the conclusion that the best predictor of 10-year survival after breast cancer surgery was the preoperative Physical Component Summary score on the SF-36. This global sensitivity analysis was also conducted with the other modes for confirmation?

Ans: Thank you for your observation. Previous studies (e.g., Tanade et al, Annu Int Conf IEEE Eng Med Biol Soc 2021;2021:4395-4398; Hosseini & Bailey, Science of The Total Environment 2022; (810): 152293; Lu & Borgonovo, Eur J Oper Res 2021 doi: 10.1016/j.ejor.2021.11.018. Online ahead of print) also conduct the global sensitivity analysis to identify the most influential potential parameters. The global sensitivity of the input variables against the output variable was expressed as the ratio of the network error (sum of squared residuals). The manuscript has been clarified accordingly.

4- With regard to the pre-processing of training data, it was not clear whether there were any missing data in the dataset and how they were treated. Nor did I see any mention of data normalization (linear or z-score?) and the possible handling of outliers.

Ans: The above issues are addressed in the subsection 2.3 Potential Predictors in the revised manuscript. Again, thank you.

5- During the Discussions, the authors claims that "One advantage of the DNN model is its capability to process incomplete or noisy inputs". This is not completely true, especially when missing data occurs in the dataset. Other models, such as those based on decision trees (e.g. XGBoost) are more robust against the occurrence of missing data.

Ans: We rewrote the sentence as “One advantage of the DNN model is its capability to process incomplete or noisy inputs more appropriately and more accurately compared to other models when no missing data occurs in the dataset.” in the Discussion section. Thank you.

6- Finally, I suggest that authors make their data and code publicly available in some repository (e.g. github) ommiting any sensitive information so that other researchers can reproduce the experiments described in the article and can conduct new experiments.

Ans: As advised by the reviewer, the revised manuscript has already mentioned that the datasets used/or analyzed during the current study are available from the corresponding author on reasonable request. Thank you for your suggestions.

Reviewer 3 Report

Please provide a table with an overview of the forecast models for better understanding of the models.

A very low number of BCT, please explain. 

DNN gives the best results, but SF36pcs is a good predictor. So what can I do to improve my results regarding survival? is there a conclusion or just a prediction?

Should DNN become commercially available?

What do the numbers in fig 3 mean, I understand that the higher the more correlation but what is the range/scale/relation?

I think the authors should offer additional ideas of the impact to improve patient care. and please write simple, as for surgeons ;-)

Author Response

Please provide a table with an overview of the forecast models for better understanding of the models.

Ans: As advised by the reviewer, the revised manuscript proved a summary of related studies in Table 1. Thank you for your suggestions.

A very low number of BCT, please explain.

Ans: Thank you for your observation. As advised, the statements in the revised Limitations section provide details of the issue as follows: “First, our study revealed the numbers of modified radical mastectomy or mastectomy with reconstruction being higher than that of breast-conserving therapy, which is contradictory to the prevalence in the USA and Europe. The process of making a treatment decision is complicated and involves many factors influencing patients’ choice of surgery type. It deserves a further study.”

DNN gives the best results, but SF36pcs is a good predictor. So what can I do to improve my results regarding survival? is there a conclusion or just a prediction?

Ans: As advised by the reviewer, the statements “The results of the current study further suggest that 10-year survival among women with breast cancer surgery could be enhanced by targeted interventions aimed at increasing patients' overall physical and mental functioning.” have been added in the Conclusions section. Again, thank you.

Should DNN become commercially available?

Ans: Thank you for your comment. Deep neural networks have recently become the standard tool for solving a variety of computer vision problems. Whereas future studies may investigate further refinements of the deep-learning DNN model applied in this study and their potential for integration with other clinical decision-making tools. Furthermore, hybrid methods may also provide additional data that can be used to improve prediction of survival after cancer surgery. Additionally, such data could also be vital for developing, promoting, and improving healthcare policies related to post-surgery treatment of cancer patients.

What do the numbers in fig 3 mean, I understand that the higher the more correlation but what is the range/scale/relation?

Ans: Thank you for your observation. Previous studies (e.g., Tanade et al, Annu Int Conf IEEE Eng Med Biol Soc 2021;2021:4395-4398; Hosseini & Bailey, Science of The Total Environment 2022; (810): 152293; Lu & Borgonovo, Eur J Oper Res 2021 doi: 10.1016/j.ejor.2021.11.018. Online ahead of print) also conduct the global sensitivity analysis to identify the most influential potential parameters. The global sensitivity of the input variables against the output variable was expressed as the ratio of the network error (sum of squared residuals). The manuscript has been clarified accordingly.

I think the authors should offer additional ideas of the impact to improve patient care. and please write simple, as for surgeons ;-)

Ans: As advised by the reviewer, the revised Conclusion section summarizes the clinical implications of the study. Again, thank you.